# Oral Candidiasis Associated with Aging and Salivary Hypofunction in Stomatitis Patients

**DOI:** 10.3390/jof11080574

**Published:** 2025-08-01

**Authors:** Yeon-Hee Lee, Solsol Seo, Tae-Seok Kim, Sang-woo Lee

**Affiliations:** 1Department of Orofacial Pain and Oral Medicine, Kyung Hee University, Kyung Hee University Dental Hospital, #613 Hoegi-dong, Dongdaemun-gu, Seoul 02447, Republic of Korea; solsolseo2@gmail.com (S.S.); taiseok11@naver.com (T.-S.K.); 2Center for Systems Biology, Massachusetts General Hospital, Harvard Medical School, 185 Cambridge Street, Boston, MA 02114, USA; 3Department of Physiology, School of Dentistry and Dental Research Institute, Seoul National University, Seoul 08826, Republic of Korea

**Keywords:** oral candidiasis, *Candida albicans*, xerostomia, salivary flow rate, age, elderly

## Abstract

**Objectives:** Stomatitis is a broad term for oral mucosal inflammation, and oral candidiasis represents one of its common subtypes caused by fungal infection. This study aimed to investigate the relationship between oral candidiasis and reduced salivary flow in patients diagnosed with stomatitis and to identify clinical predictors of oral candidiasis. **Methods:** A total of 259 patients (mean age 59.77 ± 15.93 years; range 10–87 years; 201 females) with stomatitis were evaluated for oral candidiasis through *Candida albicans* culture testing. Clinical characteristics were compared between *Candida*-positive and *Candida*-negative groups. Unstimulated salivary flow rate (UFR) and stimulated salivary flow rate (SFR) were measured to assess xerostomia. **Results:** Among the 259 patients, 81 (31.3%) were diagnosed with oral candidiasis. Patients with candidiasis were significantly older (64.25 ± 14.66 years) than those without (57.73 ± 16.10 years; *p* = 0.002). Both UFR (0.36 ± 0.32 vs. 0.47 ± 0.28 mL/min, *p* = 0.006) and SFR (1.21 ± 0.68 vs. 1.41 ± 0.69 mL/min, *p* = 0.032) were significantly lower in the candidiasis group. The prevalence of xerostomia was significantly higher among *Candida*-positive patients, based on UFR ≤ 0.2 mL/min (49.4% vs. 18.5%, *p* < 0.001) and SFR ≤ 0.7 mL/min (27.2% vs. 10.7%, *p* < 0.001). The predictive accuracy for oral candidiasis was 62.2% based on age (AUC = 0.622; cutoff 64.50 years), 65.8% for UFR (AUC = 0.658; cutoff 0.335 mL/min), and 58.7% for SFR (AUC = 0.587; cutoff 1.150 mL/min). In the generalized linear model, xerostomia, as defined by UFR, was a significant predictor of oral candidiasis (B = 0.328, 95% CI: 0.177–0.480, *p* < 0.001). **Conclusions:** Oral candidiasis in patients with stomatitis was more strongly associated with decreased UFR than with aging alone. Among the factors assessed, reduced unstimulated salivary flow may serve as a useful clinical indicator for predicting oral candidiasis, particularly in elderly individuals.

## 1. Introduction

Stomatitis is an umbrella term for inflammation of the oral mucosa, encompassing a wide range of inflammatory processes affecting the mouth and lips, regardless of the presence of oral ulcers [1]. The term is derived from the Greek word “stoma,” meaning “mouth,” and the suffix “-itis,” indicating “inflammation.” It can present with various etiologies and manifestations, including clinical conditions such as aphthous stomatitis, angular cheilitis, denture-related stomatitis, allergic contact stomatitis, migratory stomatitis, and herpetic gingivostomatitis [2]. The causes of stomatitis include infections, trauma, systemic diseases, or allergic reactions. Among these, oral candidiasis represents a distinct subtype characterized by fungal infection of the oral mucosa, most commonly caused by *Candida albicans* [3]. In this study, we focused on oral candidiasis occurring in the context of stomatitis. The prevalence of stomatitis varies widely, ranging from 19% to 79% [4]. Despite its high prevalence, clinical studies aimed at elucidating the aetio-pathophysiology and clinical characteristics of stomatitis remain insufficient.

However, the etiology of stomatitis remains unclear. Common causes of stomatitis include contamination by microorganisms, such as fungi or yeast; viral infections; xerostomia; inflammation; nutritional deficiencies in iron, folic acid, zinc, and members of the vitamin B family (riboflavin, pyridoxine, cobalamin, and niacin); poor oral hygiene; smoking; and oral trauma [5]. Additionally, allergic reactions, various medically compromised conditions, side effects following radiation therapy, diabetes mellitus, and genetic factors are associated with stomatitis [6]. Regarding SARS-CoV-2, the pathogen responsible for the coronavirus disease 2019 (COVID-19) pandemic, the most common oral symptoms are xerostomia and stomatitis, which are present in 40% of cases [7]. Xerostomia refers to the sensation of oral dryness, and stomatitis due to viral infections may occur in conjunction. Xerostomia is the quantitative reduction in saliva that exhibits anti-viral and antifungal effects. In a recent meta-analysis, xerostomia and oral candidiasis were found to be closely interrelated [8]. Consequently, individuals with xerostomia may have increased susceptibility to microbial infections. In this study, we focused on the interconnections among stomatitis, candidiasis, and xerostomia.

*Candida albicans* infections are among the most common fungal infections of the oral cavity, with *Candida* spp. being one of the leading fungal pathogens affecting humans. Oral candidiasis is an opportunistic infection affecting the oral mucosa and is caused by the increased virulence of commensal *Candida* species. *Candida albicans* is the causative agent of various forms of stomatitis, including denture-related stomatitis, angular cheilitis, and median rhomboidal glossitis [9]. *Candida albicans* is a robust, gram-positive, dimorphic yeast that can exist as a normal commensal in the oral cavity of healthy individuals and patients with stomatitis. Approximately 18% of healthy young people are carriers of *Candida albicans* [9]. The primary location of *Candida albicans* in the oral cavity is the posterior tongue and other oral sites, such as the mucosa and gingiva, while secondary colonization occurs on dental surfaces and plaque over dental prostheses [10]. The significance of orally derived *Candida albicans* extends beyond stomatitis, as it can penetrate blood vessels and cause severe brain diseases in premature infants and the elderly [11]. Particularly in the elderly, *Candida* infections are a significant and expanding clinical problem [12]. This is attributed to the prevalence of polypharmacy, reduced salivary flow rates, and a higher likelihood of frailty or concurrent systemic diseases.

Reduced salivary flow plays a pivotal role in the development of oral candidiasis by compromising the oral cavity’s natural antifungal defense. Saliva prevents *Candida albicans* from adhering to the oral epithelium through mechanical cleansing and antifungal peptides [13]. When salivary secretion decreases, this protective mechanism is impaired, promoting *Candida* adhesion and overgrowth on mucosal surfaces. *Candida albicans*, a commensal organism in the oral cavity, can readily shift to a pathogenic state under such microenvironmental changes, particularly in immunocompromised individuals or those with hyposalivation [14,15]. Oral candidiasis is classified into acute and chronic forms, including pseudomembranous, erythematous, and hyperplastic variants [16]. In addition, other *Candida*-associated conditions, such as denture stomatitis, angular cheilitis, and median rhomboid glossitis, contribute to the development of stomatitis. These conditions are frequently observed in elderly individuals, in whom age-related reductions in salivary flow and immune function predispose the oral mucosa to Candida colonization and infection.

The occurrence of stomatitis and candidiasis not only reduces the patient’s quality of life but also increases pain and discomfort in the mouth. This study hypothesized that the co-occurrence of oral candidiasis in patients with stomatitis may have significant clinical implications. Specifically, we aimed to determine whether oral candidiasis is correlated with a reduction in salivary flow rate, extension of symptom duration from onset, and an increase in subjective pain intensity. We also investigated the principal factors that could predict oral candidiasis and evaluated their predictive accuracy. Additionally, this study aimed to determine whether oral candidiasis occurs more frequently as age increases in patients with stomatitis by examining individuals across a broad age spectrum (from teens to individuals in their 80s).

## 2. Methods

Procedures involving human subjects were conducted in accordance with the ethical standards of the Committee on Human Experimentation at our institution and the 1975 Declaration of Helsinki. During the evaluation of variables, inter-rater reliability between the two examiners, T.-S.K. and Y.-H.L., was high, with Cohen’s kappa values ranging from 0.82 to 0.86. Any discrepancies between the two were resolved through multiple rounds of discussion until a consensus was reached.

### 2.1. Study Population and Study Design

This observational study included consecutive patients presenting with stomatitis at the Department of Orofacial Pain and Oral Medicine of Kyung Hee University Dental Hospital between January 2022 and May 2024. Subject selection was based on a standardized clinical examination. Stomatitis, defined as inflammation of the oral mucosa [17], includes a range of specific conditions such as aphthous stomatitis, oral candidiasis, denture stomatitis, angular cheilitis, herpes stomatitis, oral lichen planus, and lichenoid reactions. In this study, the limitations of the initial clinical examination preclude a definitive diagnosis of a specific subtype of stomatitis without further diagnostic investigations. Consequently, clinical experts rely on visual inspection and a comprehensive patient history to establish a clinical impression of stomatitis during the initial assessment, rather than assigning a definitive diagnosis to the underlying specific condition.

The male-to-female ratio of patients with stomatitis is 1:3.47. Patients were divided into two groups based on the presence or absence of oral candidiasis. Their clinical characteristics, unstimulated and stimulated salivary flow rates, presence of xerostomia based on each salivary flow rate, and prevalence of common systemic diseases were investigated and analyzed. Among the 259 patients with stomatitis, 81 patients (69 females and 12 males; mean age 64.25 ± 14.66 years) had oral candidiasis (defined as a level of a few or more in *Candida albicans* culture), while 178 patients (132 females and 46 males; mean age 57.73 ± 16.10 years) did not have oral candidiasis. Thus, we divided the patients into two groups for the analysis: stomatitis without oral candidiasis and stomatitis with oral candidiasis. We collected data and conducted various analyses to identify factors associated with oral candidiasis in the entire cohort of 259 patients, as well as differences in outcomes based on the presence or absence of oral candidiasis. Patients with insufficient data for analysis or unclear *Candida albicans* culture results were excluded.

### 2.2. Diagnosis of Oral Candidiasis

Candida cultures were performed on oral lesions by an oral medicine specialist on the first day of their visit. The area of the patient’s mouth that experienced pain or discomfort was swabbed using a sterile cotton applicator of Copan Transystem™ (COPAN Italia S.p.A., Brescia, Italy) and inoculated onto a culture dish containing Sabouraud dextrose agar. The dish was covered, sealed with paraffin wax, and incubated at 30 °C. The observation period spanned four weeks, with daily observations during the first week. From the second week onward, the intervals between observations were gradually extended to monitor the growth of *Candida albicans*. The final analysis and conclusions were drawn at the end of the fourth week. The results were categorized into four stages based on the quantity of *Candida albicans* cultured, as indicated by the area and density: none (no growth), few (covering less than half of the medium), moderate (covering more than half of the medium), and many (colonies covering most of the medium) (Figure 1). In addition to visual assessment, at the Department of Laboratory Medicine, Kyung Hee University Medical Center, oral candidiasis was confirmed through molecular microbiological analysis using a PCR assay with *Candida albicans*-specific primers (CA3: 5′-GGT TTG GAA AGA CGG TAG-3′ or CA4: 5′-AGT TTG AAG ATA TAC GTG GTA G-3′) [18]. According to our institutional experience, in over 95% of oral samples, colonies that grow on culture media are ultimately confirmed as Candida albicans through matrix-assisted laser desorption ionization time-of-flight mass spectrometry analysis. In rare cases (fewer than 5% of patients), *Candida tropicalis*, *Candida parapsilosis*, *Candida glabrata*, or *Candida krusei* may be detected, either alone or in combination with other Candida species. Based on this evidence, colonies exhibiting typical morphological features were presumed to be *C. albicans* in this study.

### 2.3. Clinical Characteristics

Subjective pain intensity was assessed using a visual analog scale (VAS). Pain was rated on an 11-point scale ranging from 0 to 10, with 0 indicating no pain and 10 indicating the worst possible pain. The patients were instructed to rate their pain accordingly, and their self-reported scores were recorded. Thus, scores closer to 10 indicated higher levels of subjective pain intensity. The duration of stomatitis symptoms in the oral cavity was recorded in months. Chronic pain was defined as pain persisting for more than six months, indicating the chronicity of the condition [19]. The lesion sites were categorized into the tongue, lips, gingiva, palate, and entire mouth based on the areas where patients reported discomfort or pain. The number of lesion sites was also determined. Oral hygiene was categorized into three levels: good, acceptable, and poor. This classification was used to analyze the relationship between oral hygiene and the presence or abundance of *Candida albicans* in patients with oral candidiasis. This study also investigated systemic diseases known to affect salivary flow rate, specifically hypertension, diabetes, osteoporosis, and cardiovascular diseases. These data were recorded based on patient reports of diagnoses or medications used for these conditions. The presence and number of systemic diseases were examined to determine their effects on oral candidiasis.

### 2.4. Measurement of Salivary Flow Rate and Diagnostic Criteria for Xerostomia

Prior to the saliva sampling session, the participants were instructed to refrain from caffeine and nicotine for at least 4 h and alcohol for at least 24 h. Whole saliva samples, both stimulated and unstimulated, were collected between 9:30 and 11:30 a.m. to minimize diurnal variability. The mean time difference between waking up and collecting was 3 h. All participants refrained from drinking alcohol the previous day and were instructed to abstain from eating, drinking, and brushing their teeth before the collection of saliva samples. The unstimulated whole salivary flow rate (UFR) was measured for 10 min using the spitting method. The stimulated whole salivary flow rate (SFR) was measured for 5 min while chewing 1 g of gum base after a 2 min pre-stimulation period to remove saliva retained in the ducts. The UFR and SFR were expressed in mL/min. The diagnostic criteria for xerostomia were set at a UFR of 0.2 mL/min and an SFR of 0.7 mL/min.

## 3. Statistical Analysis

Data were analyzed using the Statistical Package for Social Sciences (SPSS) for Windows (version 26.0; IBM Corp., Armonk, NY, USA). Descriptive statistics were presented as mean ± standard deviation or as frequencies with percentages, as appropriate. To analyze the distribution of categorical data, we employed the χ^2^ test and Bonferroni correction for equality of proportions. A *t*-test was used to compare the means between the two groups. Spearman’s correlation analysis was conducted to evaluate the strength of the association between two variables, where the absolute value of the correlation coefficient (r) indicates the strength of the correlation, ranging from −1 to +1, with values closer to 1 indicating a stronger correlation and values closer to 0 indicating a weaker correlation. Receiver operating characteristic (ROC) curves were plotted, and the corresponding area under the curve (AUC) values were calculated to assess the performance of the models at the classification threshold (above the mean value of each laboratory parameter). AUC values were interpreted as follows: AUC = 0.5 (no discrimination), 0.6 ≥ AUC > 0.5 (poor discrimination), 0.7 ≥ AUC > 0.6 (acceptable discrimination), 0.8 ≥ AUC > 0.7 (excellent discrimination), and AUC > 0.9 (outstanding discrimination). A generalized linear model was used to investigate the correlation between oral candidiasis and other factors that were either significantly correlated with oral candidiasis or showed significant differences based on oral candidiasis. Beta weights (B), standard errors, *p*-values, and 95% confidence intervals (95% CIs) were investigated. The intra-rater reliability of muscle thickness and cross-sectional area measurements was assessed using the ICC coefficient, with a mean value of 0.84. A two-tailed *p*-value of less than 0.05 was considered statistically significant for all analyses.

## 4. Results

### 4.1. Demographics

A total of 259 patients (58 males and 201 females; mean age: 59.77 ± 15.93 years; range: 10–87 years) were included. Of the 259 patients with stomatitis, 81 (31.27%) had oral candidiasis based on *Candida albicans* cultures. Regarding age, the stomatitis with oral candidiasis group (64.25 ± 14.66 years) was significantly older than the stomatitis without oral candidiasis group (57.73 ± 16.10 years) (*p* = 0.002). The proportion of females was significantly higher in the stomatitis with oral candidiasis group (male-to-female ratio = 1:5.75) than in the stomatitis without oral candidiasis group (male-to-female ratio = 1:2.87) (*p* = 0.048) (Table 1).

### 4.2. Pain, Lesion Distribution, and Oral Hygiene

The VAS for subjective pain intensity was higher in patients with stomatitis and oral candidiasis than those without, but the difference was not statistically significant (mean 3.89 ± 2.82 vs. 3.47 ± 3.07, *p* = 0.279). Contrary to expectations, the symptom duration was significantly shorter in patients with stomatitis with oral candidiasis than those without (16.39 ± 30.45 months vs. 23.56 ± 36.72 months, *p* = 0.033). Additionally, the proportion of patients with symptoms of ‘chronicity’ was lower in the stomatitis with oral candidiasis group than in the stomatitis without oral candidiasis group (39.5% vs. 52.8%, *p* = 0.047). Further studies are needed to determine whether *Candida albicans* causes pain that prompts patients with stomatitis to seek medical attention sooner.

Regarding lesion sites, the most commonly reported discomfort was the tongue (stomatitis without oral candidiasis, 55.6%; stomatitis with oral candidiasis, 53.1%; *p* = 0.788). In stomatitis patients without oral candidiasis, the lesion sites were most frequently located on the tongue (55.6%), entire mouth (38.2%), buccal mucosa (9.6%), lips (4.5%), palate (3.9%), and gingiva (3.4%). In patients with stomatitis and oral candidiasis, the lesions were most frequently located on the tongue (53.1%), the entire mouth (34.6%), the lips (14.8%), the buccal mucosa (13.6%), the gingiva (3.7%), and the palate (1.2%). The lips were the only site where lesions were more frequently found in patients with oral candidiasis than those without (4.5% vs. 14.8%, *p* = 0.01).

Over 90% of the patients with stomatitis have acceptable to good levels of oral hygiene. There was no significant difference in the distribution of good, acceptable, and poor levels between the two groups (*p* = 0.547). Both groups had acceptable levels of oral hygiene (>70%). Good oral hygiene was observed in 21.9% of the patients with stomatitis only and in 16.0% of the patients with both stomatitis and oral candidiasis. Poor oral hygiene was observed in 7.3% of patients with stomatitis only and 7.4% of patients with both stomatitis and oral candidiasis (Table 1).

### 4.3. Salivary Flow Rate and Xerostomia

Regarding salivary flow rates, the UFR and SFR were significantly lower in patients with oral candidiasis than in those without. UFR was significantly lower in the stomatitis with oral candidiasis group (0.36 ± 0.32 mL/min) compared with the stomatitis without oral candidiasis group (0.47 ± 0.28 mL/min) (*p* = 0.006). Similarly, SFR was significantly lower in the stomatitis with oral candidiasis group (1.21 ± 0.68 mL/min) compared with the stomatitis without oral candidiasis group (1.41 ± 0.69 mL/min) (*p* = 0.032). For reference, the UFR of all stomatitis patients was 0.43 ± 0.29 mL/min, and the SFR was 1.34 ± 0.69 mL/min. The proportion of xerostomia based on UFR (xerostomia in the UFR) was significantly higher in the stomatitis with oral candidiasis group (49.4%) than in the stomatitis without oral candidiasis group (18.5%) (*p* < 0.001). Similarly, the proportion of xerostomia based on SFR (xerostomia_SFR) was significantly higher in the stomatitis with oral candidiasis group (27.2%) than in the stomatitis without oral candidiasis group (10.7%) (*p* < 0.001) (Table 2).

### 4.4. Systemic Diseases

No significant differences were observed in the distribution of systemic diseases according to the presence of oral candidiasis. In the stomatitis without oral candidiasis group, the distribution was as follows: hypertension (37.1%), diabetes (11.8%), cardiovascular disease (7.9%), and osteoporosis (5.1%). In the stomatitis with oral candidiasis group, the distributions were as follows: hypertension (37.0%), diabetes mellitus (13.6%), cardiovascular disease (3.7%), and osteoporosis (2.5%). There was also no significant difference in the number of systemic diseases between the two groups (*p* = 0.619) (Table 2).

### 4.5. Factors Associated with Oral Candidiasis

When considering oral candidiasis in all patients, an increase in *Candida albicans* significantly correlated with advancing age (r = 0.219, *p* < 0.01). Additionally, there were significant positive correlations between age and the number of systemic diseases (r = 0.471, *p* < 0.001) and poor oral hygiene (r = 0.293, *p* < 0.001). The UFR and SFR were also significantly positively correlated with each other (r = 0.488, *p* < 0.001). Interestingly, in cases of oral candidiasis, the increase in the amount of *Candida albicans* was significantly correlated with a decrease in UFR (r = −0.265, *p* < 0.001) and SFR (r = −0.150, *p* = 0.016) (Figure 2). This indicates that a decrease in salivary flow rate and the presence of xerostomia are associated with increased *Candida albicans*.

### 4.6. The Cutoff Value for Oral Candidiasis

To predict the presence of oral candidiasis, we performed ROC curve analysis using age, UFR, and SFR (which showed the most significant differences based on oral candidiasis) as independent variables and oral candidiasis as the dependent variable. The prediction accuracy for oral candidiasis based on age was 62.2% (AUC = 0.622, 95% CI: 0.547–0.696, *p* = 0.002), with a cutoff value of 64.50 years for age. The prediction accuracy based on the UFR was 65.8% (AUC = 0.658, 95% CI: 0.582–0.734, *p* < 0.001), with a cutoff value of 0.335 mL/min for UFR. The prediction accuracy based on SFR was 58.7% (AUC = 0.587, 95% CI: 0.510–0.663, *p* = 0.025), with a cutoff value of 1.150 mL/min for SFR. Thus, the predictive accuracy for oral candidiasis based on the UFR was 7.1% higher than that based on the SFR (Figure 3).

### 4.7. Distribution of Oral Candidiasis and Xerostomia by Age

The incidence of oral candidiasis significantly increased in the 80s age group (22.2%) compared with that in the 10s (22.2%), 20s (0.0%), and 30s (20.0%) age groups (*p* = 0.047) (Figure 4). When examining the quantity of *Candida albicans* more closely, the proportion of moderate (22.2%) and high (33.3%) *Candida albicans* was higher in the 80s age group. In contrast, the distribution of moderate to many *Candida albicans* in the 10s–30s age groups was nearly 0.0%. However, there was no significant difference in the overall distribution of the four categories of *Candida albicans* (none, a few, moderate, and many) across different age groups (*p* = 0.238).

When examining the distribution of xerostomia by age group, the xerostomia_UFR, based on a threshold of 0.2 mL/min for UFR, showed a statistically significant difference (*p* = 0.044). The xerostomia _SFR, based on a threshold of 0.7 mL/min, is also differently distributed by age groups (*p* = 0.005) (Figure 4). The presence of xerostomia, based on the UFR and SFR, increased with age. The prevalence of Xerostomia_UFR was 11.1% in the 10s, 0.0% in the 20s, 20.0% in the 30s, 23.3% in the 40s, 27.1% in the 50s, 27.9% in the 60s, and 29.2% in the 70s, showing a significant increase to 61.1% in the 80s. The prevalence of Xerostomia_SFR was 0.0% among those in their 10s and 20s, 10.0% among those in their 30s, 20.0% among those in their 40s, 15.3% among those in their 50s, 13.1% among those in their 60s, and 12.3% among those in their 70s, with a significant increase to 50% in the 80s. More than half of stomatitis patients aged >80 years had xerostomia (Table 3).

### 4.8. Generalized Linear Model for Oral Candidiasis

When a generalized linear model was performed to predict oral candidiasis, significant predictors were xerostomia_UFR (B = 0.328, 95% CI: 0.177–0.480, *p* < 0.001) and lesions on the lips (B = 0.280, 95% CI: 0.085–0.475, *p* < 0.005). Age (B = 0.001, 95% PI: −0.017–0.019, *p* = 0.907) was not a significant predictor of oral candidiasis when considered alongside the UFR. Thus, when multiple parameters were included in the analysis, only xerostomia based on UFR and lip lesions significantly predicted oral candidiasis. This suggests that the salivary flow rate in a neutral state without chewing stimulation is most closely associated with the occurrence of oral candidiasis (Table 4).

## 5. Discussion

Approximately one-third of the patients with stomatitis had oral candidiasis. Patients with oral candidiasis were significantly older than those without candidiasis. The unstimulated and stimulated salivary flow rates were significantly lower in patients with oral candidiasis. The proportion of patients with xerostomia, based on both unstimulated and stimulated salivary flow rates, was significantly higher in the oral candidiasis group. Among the predictors assessed (age, UFR, and SFR), UFR showed the highest predictive accuracy for oral candidiasis. In the generalized linear model, xerostomia defined by UFR was identified as a significant predictor. Additionally, with increasing age, the occurrence of xerostomia based on UFR and oral candidiasis also increased. Contrary to our expectations, age alone was not a significant factor for oral candidiasis, and there was no difference in subjective pain intensity between patients with and without candidiasis. In addition, denture stomatitis is often asymptomatic and cannot be diagnosed based on subjective symptoms alone [20], highlighting the importance of objective assessments such as clinical examination, salivary flow measurements, and mycological testing for identifying oral candidiasis, particularly in denture wearers. Taken together, these findings underscore the importance of considering salivary gland function, aging, and systemic factors in the evaluation and management of oral candidiasis and stomatitis.

First, the oral cavity serves as a marker area that reflects the health of the oral cavity and the entire body. It is often considered a window to the body because oral manifestations can indicate many systemic diseases or conditions. The first line of defense against microbial infection or inflammation of the oral cavity, as well as physical friction and chemical stimulation, includes saliva and oral mucosa, which are essential for the human body [21]. The average age of the stomatitis patients in this study was relatively high (59.77 years), with the prevalence of xerostomia and candidiasis increasing with age. Major aphthous stomatitis predominantly occurs in individuals aged 35–59, whereas minor aphthous stomatitis is more common in individuals aged <30 [6]. Although the precise prevalence of stomatitis is not well-documented, it has been theoretically posited that the likelihood of developing oral diseases increases with age.

As age increases, the likelihood of developing oral mucosal diseases increases due to decreased saliva secretion, difficulties in maintaining oral hygiene, the presence of dental prostheses, smoking, alcohol consumption, polypharmacy, and an increase in systemic diseases [22]. The salivary flow rate of the submandibular and sublingual salivary glands decreases with age. Still, the parotid gland, responsible for 90% of saliva production, shows no significant change in salivary flow rate with age [23]. Xerostomia can occur at any age; however, approximately 25% of the elderly suffer from oral dryness and related complaints [24]. The decrease in salivary flow rate with increasing age has not yet led to consistent conclusions owing to the comorbidities and complexities associated with systemic diseases in the elderly. Regarding the oral mucosa, even when it appears clinically normal, aging can cause changes, such as a reduction in the thickness of the epithelial and keratinized layers [25]. The mucosa becomes thinner, smoother, more fragile, and more susceptible to infection by *Candida albicans* and other microorganisms. Consequently, they are more prone to damage and exhibit slower healing [26]. Therefore, in humans, the reduction in saliva, stomatitis, and oral candidiasis can be closely related, with aging being a significant factor to consider in these relationships.

Among the various analytical methods, UFR has consistently emerged as a predictor of oral candidiasis in patients with stomatitis. The cutoff value of UFR for predicting oral candidiasis was 0.335 mL/min. Generally, the normal range for UFR is 0.3–0.4 mL/min, and xerostomia is diagnosed when UFR is below 0.1–0.2 mL/min [24]. In a recent meta-analysis, xerostomic patients had a threefold higher risk of oral *Candida* growth and candidiasis development than controls [8]. In this study, many patients experienced xerostomia (28.2% xerostomia_UFR and 15.8% xerostomia_SFR), and the UFR had a higher weight on xerostomia than the SFR. An adequate amount of unstimulated saliva is crucial for oral health because it contains various antimicrobial constituents, antibodies, enzymes, hormones, growth factors, lubricants, and water. These components enable the saliva to perform various functions, including antifungal action, immunological protection, and lubrication. However, when employing advanced machine learning techniques, the absolute salivary flow rate that induces xerostomia varies according to the patient’s age, sex, and the number of systemic diseases [27]. Decreased unstimulated or residual saliva is associated with oral dryness [28]. Additionally, the microbiome profile differs, and species diversity is lower in unstimulated saliva than in stimulated saliva [29]. However, further studies with larger sample sizes are required to examine the compositional differences between unstimulated and stimulated saliva. The key factors causing the quantitative and qualitative differences between saliva under these two conditions remain unknown.

We focused on the relationship between the growth of *Candida albicans* and the presence of oral candidiasis in patients with stomatitis. However, there is a significant lack of research specifically addressing the role of *C. albicans* in the context of stomatitis. Oral candidiasis is a common opportunistic infection of the oral cavity caused by the overgrowth of *Candida* species, with *C. albicans* being the most prevalent [30]. Several local factors contribute to the development of oral candidiasis, including impaired salivary gland function, xerostomia, and poor hygiene of dental prostheses. In addition, unsuitable or old removable appliances and resin-based materials can further facilitate microbial colonization, creating an environment conducive to fungal overgrowth [31]. The tongue is also considered a major site for stomatitis-related symptoms, regardless of the presence of oral candidiasis, as it provides a favorable habitat for microorganisms [32]. Accumulated metabolic debris, such as food particles and desquamated epithelial cells, along with environmental factors such as humidity and temperature, further promote microbial proliferation on the tongue surface [33]. Beyond these local factors, systemic conditions also play a critical role. Antibiotic use, certain medications, aging, malnutrition, and immunosuppression are well-known systemic contributors to the development of oral candidiasis [34].

The prevalence of oral candidiasis in the elderly ranges from 13 to 47% [34]. In this study, oral candidiasis occurred in 31.3% of patients with stomatitis, with a steep increase observed in 27.9% of patients in their 60s, 38.5% of those in their 70s, and 61.1% of those in their 80s. Individuals who are extremely old or have an immature or weakened immune system are particularly susceptible to candidiasis [35]. Oral carriage of *Candida* spp. is generally found in 30–45% of the healthy adult population [30]. In this study, the age group in which *Candida* spp. occurred beyond the normal range (up to 45%) was >80 years. The cutoff value for predicting oral candidiasis in patients with stomatitis was 64.50 years. To clarify our findings and link them with previous research, the influence of microorganisms residing on the tongue coating or in the fissures of the tongue on oral stomatitis warrants further investigation in neonates and infants with immature immune systems as well as in elderly patients with compromised or fragile immune systems.

*Candida* spp. leave the oral cavity in the bloodstream and move to various organs in the body. Overgrowth of *Candida* spp. in the oral cavity may lead to their dissemination to distant organs, potentially having clinical implications at a distance as well. Systemic candidiasis is less frequent than oral candidiasis but has a mortality rate of 71–79% [36]. Unlike young and healthy individuals, *Candida albicans* can cause fatal outcomes in frail and debilitated elderly patients, necessitating prompt diagnosis and treatment. This study was conducted during the ongoing COVID-19 pandemic. During this period, several cases of oral candidiasis in SARS-CoV-2-positive individuals have been reported [37]. The immune-inflammatory hypo-reactions and immunosuppression found in individuals with COVID-19 could favor the proliferation of *Candida* species and subsequent infections. Therefore, to prevent the sequelae of systemic candidiasis, prompt management and control of oral candidiasis is imperative.

Further research is needed to understand the relationship between *Candida albicans* and other microbes, their role in the overall microbiome profile, and the systemic and oral health conditions of the host. This is particularly important because recent studies have demonstrated that alterations in oral diseases are not solely attributed to changes in a single pathogenic species [38,39]. Rather, such alterations arise from complex shifts in the overall microbial community structure, which are significantly influenced by host-related factors including aging, salivary gland dysfunction, and immune status. These host factors dynamically interact with the microbiome, leading to disruptions in the ecological balance of the oral cavity [40]. Furthermore, the close interrelationship among aging, decreased salivary flow, and oral candidiasis underscores the importance of considering salivary gland function as a central factor in the prevention and management of stomatitis in elderly populations. Given that saliva plays a pivotal role in maintaining oral homeostasis through antifungal, antibacterial, and immunological mechanisms [41,42], its reduction with age creates an environment conducive to *Candida* colonization and subsequent infection. This highlights the need for early identification and management of xerostomia to mitigate the risk of oral candidiasis in patients with stomatitis.

Despite the strengths of this study, several limitations should be noted. The uneven distribution of patients across different age groups, despite the large sample size of over 250 participants, may have introduced bias in age-related analyses. In addition, data on oral health status, such as the number of teeth, presence of prostheses, caries, and periodontal disease, were not collected in this study. Future studies with a larger number of participants evenly distributed across age groups are warranted to address this issue. Additionally, this study did not comprehensively examine the impact of oral candidiasis on pain intensity, symptom chronicity, or overall quality of life in patients with stomatitis. As stomatitis refers to a condition characterized by inflammation and pain in the oral cavity rather than a specific disease, additional investigations are necessary to further elucidate its underlying causes. Advanced and sophisticated analytical methods should also be employed to identify individual factors contributing to the occurrence of oral candidiasis in patients with stomatitis and to clarify the interrelationships among these factors. Moreover, a multicenter prospective study with healthy controls, aligned with the concept and perspective of this research, would provide more robust evidence to support our findings.

## 6. Conclusions

A notable strength of this study is the scientific demonstration that oral candidiasis in patients with stomatitis has significant clinical implications. In patients with stomatitis, oral candidiasis was associated with decreased unstimulated salivary flow rate and increasing age. Specifically, the cutoff value for predicting oral candidiasis based on unstimulated salivary flow was 0.335 mL/min. Additionally, the prevalence of oral candidiasis increased significantly with age, with a cutoff value of 64.50 years for age. It should be noted that this age-related trend may not apply uniformly to all types of stomatitis. In particular, the epidemiology of denture stomatitis has been reported to differ, with acute symptoms often observed in younger individuals, while mucosal fibrosis in older patients may offer some protective effect against fungal colonization. Therefore, the association between aging and oral candidiasis should be interpreted in the context of salivary function and other contributing factors, rather than age alone. Another strength of this study is the validation of our hypotheses across a broad age range, from teenagers to individuals in their 80s. Expanding on this study, utilizing advanced oral microbiome analyses to identify specific profiles of stomatitis patients and microorganisms related to *Candida albicans* will be instrumental in furthering our understanding of this disease.

## Figures and Tables

**Figure 1 jof-11-00574-f001:**
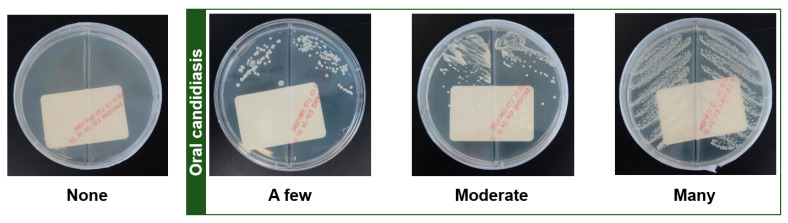
Quantity of *Candida albicans* cultured at four weeks and oral candidiasis.

**Figure 2 jof-11-00574-f002:**
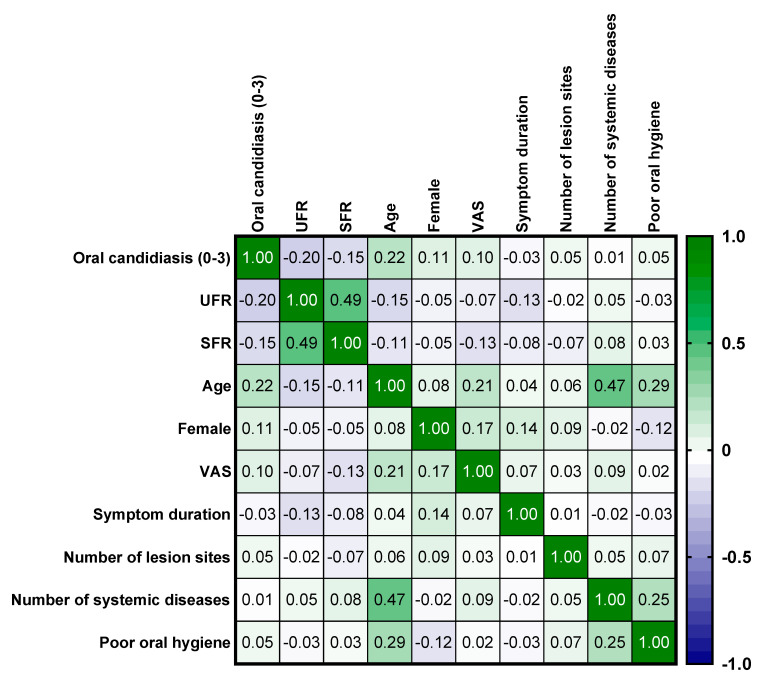
Heatmap of oral candidiasis. Results were obtained using Spearman’s correlation analysis. UFR, unstimulated salivary flow rate; SFR, stimulated salivary flow rate; VAS, visual analog scale.

**Figure 3 jof-11-00574-f003:**
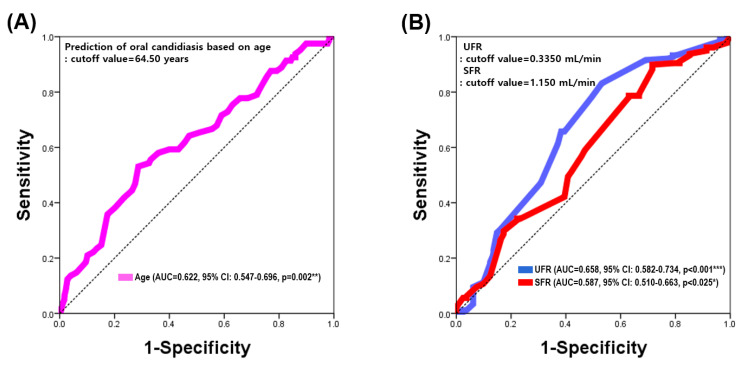
Prediction of oral candidiasis based on age and salivary flow rate. (**A**) Prediction of presence of oral candidiasis based on age; (**B**) prediction of the absence of oral candidiasis based on UFR and SFR. Results were obtained using receiver operating characteristic (ROC) analysis. UFR, unstimulated flow rate; SFR, stimulated salivary flow rate; CI, confidence interval. Statistical significance was set at *p* < 0.05. * *p* < 0.05; ** *p* < 0.01; *** *p* < 0.001.

**Figure 4 jof-11-00574-f004:**
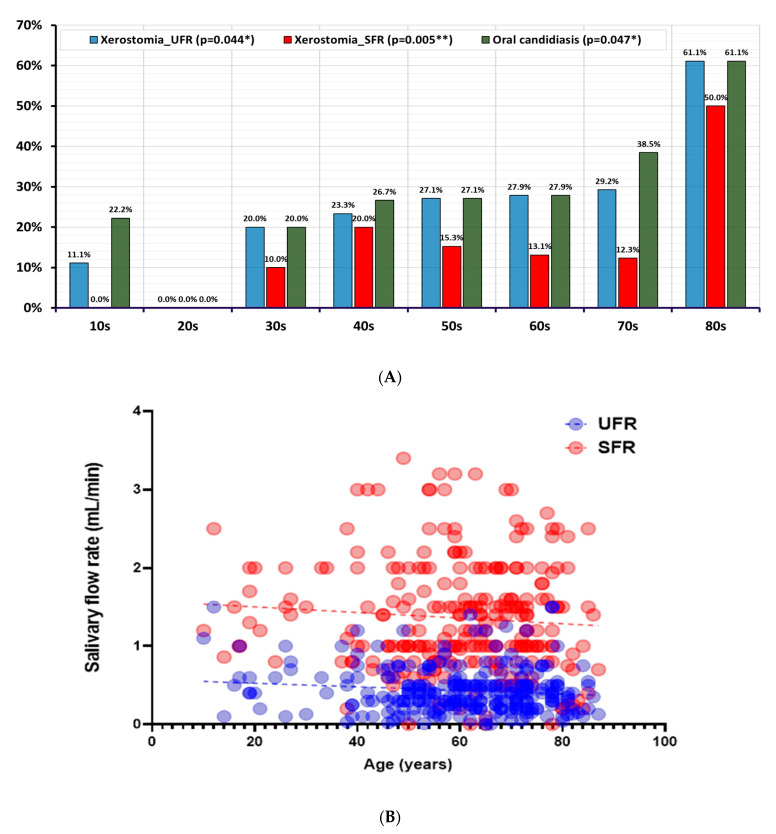
Increase in xerostomia and oral candidiasis and decrease in salivary flow rate with age. UFR: unstimulated flow rate, SFR: stimulated salivary flow rate. (**A**) Xerostomia_UFR, xerostomia_SFR, and oral candidiasis; (**B**) salivary flow rate with increasing age. In (**A**), Chi-square tests with Bonferroni correction were used to compare age-group proportions. In (**B**), Spearman’s correlation test was used to assess the association between age and salivary flow rate. Statistical significance was set at *p* < 0.05. * *p* < 0.05; ** *p* < 0.01.

**Table 1 jof-11-00574-t001:** Demographics and clinical characteristics.

	Stomatitis Without Oral Candidiasis (n = 178)	Stomatitis with Oral Candidiasis (n = 81)	*p*-Value
Demographics			
Age (years) ^a^	57.73 ± 16.10	64.25 ± 14.66	0.002 **
Sex ^b^			
Male	46 (25.8%)	12 (14.8%)	0.048 *
Female	132 (74.2%)	69 (85.2%)	
Clinical symptoms			
VAS (0–10) ^a^	3.47 ± 3.07	3.89 ± 2.82	0.279
Symptom duration (months) ^a^	16.39 ± 30.45	12.65 ± 35.12	0.033 *
Chronicity ^b^	94 (52.8%)	32 (39.5%)	0.047 *
Lesion site			
Tongue ^b^	99 (55.6%)	43 (53.1%)	0.788
Buccal mucosa ^b^	17 (9.6%)	11 (13.6%)	0.333
Lips ^b^	8 (4.5%)	12 (14.8%)	0.010 *
Gingiva ^b^	6 (3.4%)	3 (3.7%)	1.000
Palate ^b^	7 (3.9%)	1 (1.2%)	0.245
Entire mouth ^b^	68 (38.2%)	28 (34.6%)	0.575
Number of lesion sites ^b^	1.12 ± 0.47	1.17 ± 0.52	0.418
Oral hygiene ^b^			
Good	39 (21.9%)	13 (16.0%)	0.547
Acceptable	126 (70.8%)	62 (76.5%)	
Poor	13 (7.3%)	6 (7.4%)	

^a^ Results were analyzed using *t*-tests. ^b^ Results were obtained using χ^2^ test between two age groups. VAS: visual analogue scale. *p*-value significance was set at <0.05. * *p*-value < 0.05, ** *p*-value < 0.01.

**Table 2 jof-11-00574-t002:** Salivary flow rate, xerostomia, and systemic diseases.

	Stomatitis Without Oral Candidiasis (n = 178)	Stomatitis with Oral Candidiasis (n = 81)	*p*-Value
Salivary flow rate			
UFR (mL/min) ^a^	0.47 ± 0.28	0.36 ± 0.32	0.006 **
SFR (mL/min) ^a^	1.41 ± 0.69	1.21 ± 0.68	0.032 *
Xerostomia_UFR ^b^	33 (18.5%)	40 (49.4%)	<0.001 ***
Xerostomia_SFR ^b^	19 (10.7%)	22 (27.2%)	0.002 **
Systemic diseases			
Hypertension ^b^	66 (37.1%)	30 (37.0%)	1.000
Diabetes mellitus ^b^	21 (11.8%)	11 (13.6%)	0.687
Osteoporosis ^b^	9 (5.1%)	2 (2.5%)	0.511
Cardiovascular diseases ^b^	14 (7.9%)	3 (3.7%)	0.283
Number of systemic diseases ^a^	0.62 ± 0.84	0.57 ± 0.71	0.619

^a^ Results were analyzed using *t*-tests. ^b^ Results were obtained using χ^2^ test between two age groups. UFR: unstimulated salivary flow rate, SFR: stimulated salivary flow rate. *p*-value significance was set at <0.05. * *p*-value < 0.05, ** *p*-value < 0.01, *** *p*-value < 0.001.

**Table 3 jof-11-00574-t003:** Distribution of oral candidiasis and xerostomia according to age.

	10s (n = 9)	20s (n = 7)	30s (n = 10)	40s (n = 30)	50s (n = 59)	60s (n = 61)	70s (n = 65)	80s (n = 18)	*p*-Value
Proportion of oral candidiasis									
Oral candidiasis	2 (22.2%)	0 (0.0%)	2 (20.0%)	8 (26.7%)	16 (27.1%)	17 (27.9%)	25 (38.5%)	11 (61.1%)	0.047 *
Quantity of Candida albicans									
None	7 (77.8%)	6 (85.7%)	8 (0.8%)	22 (73.3%)	43 (72.9%)	44 (72.1%)	40 (61.5%)	7 (38.9%)	0.238
A few	2 (22.2%)	1 (14.3%)	1 (0.1%)	3 (0.1%)	7 (11.9%)	7 (11.5%)	10 (15.4%)	1 (5.6%)	
Moderate	0 (0.0%)	0 (0.0%)	1 (0.1%)	2 (6.7%)	5 (8.5%)	3 (4.9%)	5 (7.7%)	4 (22.2%)	
Many	0 (0.0%)	0 (0.0%)	0 (0.0%)	3 (10.0%)	4 (6.8%)	7 (11.5%)	10 (15.4%)	6 (33.3%)	
Xerostomia									
Xerostomia_UFR	1 (11.1%)	0 (0.0%)	2 (20.0%)	7 (23.3%)	18 (27.1%)	17 (27.9%)	19 (29.2%)	11 (61.1%)	0.044 *
Xerostomia_SFR	0 (0.0%)	0 (0.0%)	1 (10.0%)	6 (20.0%)	9 (15.3%)	8 (13.1%)	8 (12.3%)	9 (50.0%)	0.005 **

Results were obtained using repeated χ^2^ test between age groups. UFR: unstimulated salivary flow rate; SFR: stimulated salivary flow rate. Statistical significance was set at *p* < 0.05. * *p* < 0.05; ** *p* < 0.01.

**Table 4 jof-11-00574-t004:** Generalized linear model for predicting oral candidiasis.

**Parameter**	**B**	**SE**	**95% CI Lower**	**95% CI Upper**	***p*-Value**
Female [ref. = male]	0.110	0.065	−0.017	0.237	0.090
Xerostomia_UFR [ref. = none]	0.328	0.077	0.177	0.480	<0.001 ***
Xerostomia_SFR [ref. = none]	0.059	0.092	−0.120	0.239	0.517
Age	0.001	0.009	−0.017	0.019	0.907
Symptom duration	0.016	0.049	−0.080	0.112	0.743
Number of systemic diseases	0.001	0.007	−0.013	0.018	0.915
Lips [ref. = none]	0.280	0.099	0.085	0.475	0.005 **
Constant	0.178	0.016	0.150	0.212	0.168

The results were obtained using a generalized linear model. The likelihood ratio chi-square statistic was 49.014 with *p* < 0.001. UFR, unstimulated salivary flow rate; SFR, stimulated salivary flow rate; CI, confidence interval. *p*-value significance was set at <0.05. ** *p*-value < 0.01, *** *p*-value < 0.001.

## Data Availability

The datasets generated and/or analyzed during the current study are not publicly available due to ethical restrictions imposed by the Institutional Review Board (IRB), which approved their use exclusively for this study to protect patient privacy. Upon reasonable request and IRB approval, the data can be obtained from the corresponding authors.

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
