# Peer review of "Oral Candidiasis Associated with Aging and Salivary Hypofunction in Stomatitis Patients"

_jof, 2025, doi:10.3390/jof11080574_

Round 1

Reviewer 1 Report

Journal JoF (ISSN 2309-608X)

Manuscript ID jof-3749964

Type Article

Title Oral Candidiasis Associated with Aging and Salivary Hypofunction in Stomatitis Patients

Authors Yeon-Hee Lee * , Solsol Seo , Tae-Seok Kim , Sang-woo Lee *

This study aimed to determine whether oral candidiasis is correlated with decreased salivary flow, increased symptom duration, and increased subjective pain intensity. In addition, this study aimed to determine whether oral candidiasis is more common with age in patients with stomatitis, by examining individuals across a wide age range (from adolescence to octogenarian).     It is difficult to clearly distinguish between stomatitis and oral candidiasis, as the candidal etiology is common to both pathologies with or without decreased salivary flow. In these conditions, it is necessary to clearly define the definitions of the two pathologies.

   Stomatitis is an inflammation of the oral mucosa. It can be caused by an infection or induced by certain medications, carences or radiotherapy. There are several types: mycotic stomatitis; acute candidal stomatitis.

  While oral candidiasis is an opportunistic mycosis affecting the oral mucosa. This pathology mainly affects immuno-vulnerable populations, elderly and/or those exposed to fungal contamination. This condition is generally classified into acute and chronic forms, including pseudomembranous, erythematous and hyperplastic variants, as well as other Candida-associated diseases such as denture stomatitis, angular cheilitis and median rhomboid glossitis.

Oral candidiasis primarily involves Candida albicans and affects the oral mucosa. C. albicans, a commensal eukaryote  organism adapted to its host, can transform into a pathogen when changes in the microenvironment occur. In the oral cavity, in planktonic or mycofilm form, C. albicans and bacteria develop synergy. Since saliva's role is to prevent Candida from adhering to epithelial cells, thanks to anti-candidal peptides, a decrease in its production promotes oral candidiasis.

In order to avoid any confusion between stomatitis and oral candidiasis, it is better to clarify from the outset that oral candidiasis is a subsection of stomatitis.

Abstract :  There are several repetitions in the summary, 12 times "oral candidiasis" 8 times "stomatitis" which complicates reading. The distinction between stomatitis and oral candidiasis should be made at the outset in the abstract. 

 Conclusion: This statement: "Furthermore, oral candidiasis occurred significantly more frequently with age in patients with stomatitis." should be modulated because it is not applicable to denture stomatitis. Indeed, the epidemiology of denture stomatitis lists acute symptoms evident in young patients compared to older patients. Heightened immune defenses, poor hygiene, smoking, diet  … provide one explanation. In addition, fibrosis of the mucosal connective tissue in older patients represents a protection against denture covering.

Add recent reference like…

Tamai, R.; Kiyoura, Y. Candida Infections: The Role of Saliva in Oral Health—A Narrative Review. Microorganisms 2025, 13, 717. https://doi.org/10.3390/ microorganisms13040717

Journal JoF (ISSN 2309-608X)

Manuscript ID jof-3749964

Type Article

Title Oral Candidiasis Associated with Aging and Salivary Hypofunction in Stomatitis Patients

Authors Yeon-Hee Lee * , Solsol Seo , Tae-Seok Kim , Sang-woo Lee *

Abstract :  There are several repetitions in the summary, 12 times "oral candidiasis" 8 times "stomatitis" which complicates reading. The distinction between stomatitis and oral candidiasis should be made at the outset in the abstract.

 Line 154: The identification and quantification of C albicans by culture alone currently remains subjective. Other fungi, bacteria, and viruses may be involved. At what quantitative threshold is C albicans considered pathogenic?

 Line 171:  It is important to note that prosthetic stomatitis is mostly painless. It is only a clinical examination that allows this pathology to be diagnosed, confirmed by a mycological analysis.

 Line 246: To enhance the value of this descriptive study, it would have been desirable to have more details on the number of teeth present, the presence of a removable or fixed prosthesis, caries or periodontal disease.

 Line 335 :   This sentence requires a bibliographic reference. In these conditions, do you consider that oral candidiasis is part of stomatitis? 

Line 409 : addition to undisinfected prostheses, there are unsuitable, old removable appliances and resinous materials that facilitate microbial colonization. 

 Conclusion: This statement: "Furthermore, oral candidiasis occurred significantly more frequently with age in patients with stomatitis." should be modulated because it is not applicable to denture stomatitis. Indeed, the epidemiology of denture stomatitis lists acute symptoms evident in young patients compared to older patients. Heightened immune defenses, poor hygiene, smoking, diet  … provide one explanation. In addition, fibrosis of the mucosal connective tissue in older patients represents a protection against denture covering.

Add recent reference like…

Tamai, R.; Kiyoura, Y. Candida Infections: The Role of Saliva in Oral Health—A Narrative Review. Microorganisms 2025, 13, 717. https://doi.org/10.3390/ microorganisms13040717

Author Response

We sincerely appreciate your thoughtful and constructive comments. We have carefully considered each of your suggestions to improve the clarity and scientific rigor of our manuscript. Your valuable insights have significantly contributed to enhancing the overall quality of our work. All revised sections have been highlighted in red for your convenience. Once again, we would like to express our sincere gratitude for your time and efforts in reviewing our manuscript.

Major Comments
Journal JoF (ISSN 2309-608X)

Manuscript ID jof-3749964

Type Article

Title Oral Candidiasis Associated with Aging and Salivary Hypofunction in Stomatitis Patients

Authors Yeon-Hee Lee * , Solsol Seo , Tae-Seok Kim , Sang-woo Lee *

This study aimed to determine whether oral candidiasis is correlated with decreased salivary flow, increased symptom duration, and increased subjective pain intensity. In addition, this study aimed to determine whether oral candidiasis is more common with age in patients with stomatitis, by examining individuals across a wide age range (from adolescence to octogenarian).     It is difficult to clearly distinguish between stomatitis and oral candidiasis, as the candidal etiology is common to both pathologies with or without decreased salivary flow. In these conditions, it is necessary to clearly define the definitions of the two pathologies.

Stomatitis is an inflammation of the oral mucosa. It can be caused by an infection or induced by certain medications, carences or radiotherapy. There are several types: mycotic stomatitis; acute candidal stomatitis.

While oral candidiasis is an opportunistic mycosis affecting the oral mucosa. This pathology mainly affects immuno-vulnerable populations, elderly and/or those exposed to fungal contamination. This condition is generally classified into acute and chronic forms, including pseudomembranous, erythematous and hyperplastic variants, as well as other Candida-associated diseases such as denture stomatitis, angular cheilitis and median rhomboid glossitis.

Oral candidiasis primarily involves Candida albicans and affects the oral mucosa. C. albicans, a commensal eukaryote  organism adapted to its host, can transform into a pathogen when changes in the microenvironment occur. In the oral cavity, in planktonic or mycofilm form, C. albicans and bacteria develop synergy. Since saliva's role is to prevent Candida from adhering to epithelial cells, thanks to anti-candidal peptides, a decrease in its production promotes oral candidiasis.

In order to avoid any confusion between stomatitis and oral candidiasis, it is better to clarify from the outset that oral candidiasis is a subsection of stomatitis.

Response: We fully agree with your valuable suggestion and find it appropriate. Therefore, we have incorporated the following paragraphs and references into the Introduction and Discussion sections to address this point.

INTRODUCTION

Stomatitis is an umbrella term for inflammation of the oral mucosa, encompassing a wide range of inflammatory processes affecting the mouth and lips, regardless of the presence of oral ulcers [1]. The term is derived from the Greek word "stoma," meaning "mouth," and the suffix "-itis," indicating "inflammation." It can present with various etiologies and manifestations, including clinical conditions such as aphthous stomatitis, angular cheilitis, denture-related stomatitis, allergic contact stomatitis, migratory stomatitis, and herpetic gingivostomatitis [2]. The causes of stomatitis include infections, trauma, systemic diseases, or allergic reactions. Among these, oral candidiasis represents a distinct subtype characterized by fungal infection of the oral mucosa, most commonly caused by Candida albicans [3]. In this study, we focused on oral candidiasis occurring in the context of stomatitis. The prevalence of stomatitis varies widely, ranging from 19% to 79% [4]. Despite its high prevalence, clinical studies aimed at elucidating the aetio-pathophysiology and clinical characteristics of stomatitis remain insufficient.

--

Reduced salivary flow plays a pivotal role in the development of oral candidiasis by compromising the oral cavity's natural antifungal defense. Saliva prevents Candida albicans from adhering to the oral epithelium through mechanical cleansing and antifungal peptides [13]. When salivary secretion decreases, this protective mechanism is impaired, promoting Candida adhesion and overgrowth on mucosal surfaces. Candida albicans, a commensal organism in the oral cavity, can readily shift to a pathogenic state under such microenvironmental changes, particularly in immunocompromised individuals or those with hyposalivation [14,15]. Oral candidiasis is classified into acute and chronic forms, including pseudomembranous, erythematous, and hyperplastic variants [16]. In addition, other Candida-associated conditions, such as denture stomatitis, angular cheilitis, and median rhomboid glossitis, contribute to the development of stomatitis. These conditions are frequently observed in elderly individuals, in whom age-related reductions in salivary flow and immune function predispose the oral mucosa to Candida colonization and infection.

DISCUSSION

Further research is needed to understand the relationship between Candida albicans and other microbes, their role in the overall microbiome profile, and the systemic and oral health conditions of the host. This is particularly important because recent studies have demonstrated that alterations in oral diseases are not solely attributed to changes in a single pathogenic species [35,36]. Rather, such alterations arise from complex shifts in the overall microbial community structure, which are significantly influenced by host-related factors including aging, salivary gland dysfunction, and immune status. These host factors dynamically interact with the microbiome, leading to disruptions in the ecological balance of the oral cavity [37]. Furthermore, the close interrelationship among aging, decreased salivary flow, and oral candidiasis underscores the importance of considering salivary gland function as a central factor in the prevention and management of stomatitis in elderly populations. Given that saliva plays a pivotal role in maintaining oral homeostasis through antifungal, antibacterial, and immunological mechanisms [38], its reduction with age creates an environment conducive to Candida colonization and subsequent infection. This highlights the need for early identification and management of xerostomia to mitigate the risk of oral candidiasis in patients with stomatitis.

Abstract :  There are several repetitions in the summary, 12 times "oral candidiasis" 8 times "stomatitis" which complicates reading. The distinction between stomatitis and oral candidiasis should be made at the outset in the abstract. 

Response: Thank you for your valuable comment. As you pointed out, the terms "oral candidiasis" and "stomatitis" were unnecessarily repeated throughout the abstract, which affected the clarity and readability of the text. We have carefully revised the abstract to eliminate redundancy and improve its readability. In addition, we clarified at the outset that stomatitis is a broad term for oral mucosal inflammation, and oral candidiasis represents a specific subtype of stomatitis caused by fungal infection.

The revised abstract is as follows:

Objectives:

Stomatitis is a broad term for oral mucosal inflammation, and oral candidiasis represents one of its common subtypes caused by fungal infection. This study aimed to investigate the relationship between oral candidiasis and reduced salivary flow in patients diagnosed with stomatitis and to identify clinical predictors of oral candidiasis.

Methods:

A total of 259 patients (mean age 59.77 ± 15.93 years; range 10–87 years; 201 females) with stomatitis were evaluated for oral candidiasis through Candida albicans culture testing. Clinical characteristics were compared between Candida-positive and Candida-negative groups. Unstimulated salivary flow rate (UFR) and stimulated salivary flow rate (SFR) were measured to assess xerostomia.

Results:

Among the 259 patients, 81 (31.3%) were diagnosed with oral candidiasis. Patients with candidiasis were significantly older (64.25 ± 14.66 years) than those without (57.73 ± 16.10 years; p = 0.002). Both UFR (0.36 ± 0.32 vs. 0.47 ± 0.28 mL/min, p = 0.006) and SFR (1.21 ± 0.68 vs. 1.41 ± 0.69 mL/min, p = 0.032) were significantly lower in the candidiasis group. The prevalence of xerostomia was significantly higher among Candida-positive patients, based on UFR ≤ 0.2 mL/min (49.4% vs. 18.5%, p < 0.001) and SFR ≤ 0.7 mL/min (27.2% vs. 10.7%, p < 0.001). The predictive accuracy for oral candidiasis was 62.2% based on age (AUC = 0.622; cutoff 64.50 years), 65.8% for UFR (AUC = 0.658; cutoff 0.335 mL/min), and 58.7% for SFR (AUC = 0.587; cutoff 1.150 mL/min). In the generalized linear model, xerostomia, as defined by UFR, was a significant predictor of oral candidiasis (B = 0.328, 95% CI: 0.177–0.480, p < 0.001).

Conclusions:

Oral candidiasis in patients with stomatitis was more strongly associated with decreased UFR than with aging alone. Among the factors assessed, reduced unstimulated salivary flow may serve as a useful clinical indicator for predicting oral candidiasis, particularly in elderly individuals.

Conclusion: This statement: "Furthermore, oral candidiasis occurred significantly more frequently with age in patients with stomatitis." should be modulated because it is not applicable to denture stomatitis. Indeed, the epidemiology of denture stomatitis lists acute symptoms evident in young patients compared to older patients. Heightened immune defenses, poor hygiene, smoking, diet  … provide one explanation. In addition, fibrosis of the mucosal connective tissue in older patients represents a protection against denture covering.

Response: We fully agree with your valuable comment. As you pointed out, the association between oral candidiasis and aging cannot be generalized to all types of stomatitis, particularly denture stomatitis. We have carefully revised the Conclusion to clarify this point and to reflect that the relationship between aging and oral candidiasis should be interpreted in the context of salivary function and other contributing factors, rather than age alone.

Accordingly, the Conclusion has been revised as follows:

Conclusion

A notable strength of this study is the scientific demonstration that oral candidiasis in patients with stomatitis has significant clinical implications. In patients with stomatitis, oral candidiasis was associated with decreased unstimulated salivary flow rate and increasing age. Specifically, the cut-off value for predicting oral candidiasis based on unstimulated salivary flow was 0.335 mL/min. Additionally, the prevalence of oral candidiasis increased significantly with age, with a cut-off value of 64.50 years for age. However, it should be noted that this age-related trend may not apply uniformly to all types of stomatitis. In particular, the epidemiology of denture stomatitis has been reported to differ, with acute symptoms often observed in younger individuals, while mucosal fibrosis in older patients may offer some protective effect against fungal colonization. Therefore, the association between aging and oral candidiasis should be interpreted in the context of salivary function and other contributing factors, rather than age alone. Another strength of this study is the validation of our hypotheses across a broad age range, from teenagers to individuals in their 80s. Expanding on this study, utilizing advanced oral microbiome analyses to identify specific profiles of stomatitis patients and microorganisms related to Candida albicans will be instrumental in furthering our understanding of this disease.

Add recent reference like…

Tamai, R.; Kiyoura, Y. Candida Infections: The Role of Saliva in Oral Health—A Narrative Review. Microorganisms 2025, 13, 717. https://doi.org/10.3390/ microorganisms13040717

Response: We have additionally cited relevant literature to further support and strengthen our study. In particular, the article you kindly recommended has been cited as follows:

Given that saliva plays a pivotal role in maintaining oral homeostasis through antifungal, antibacterial, and immunological mechanisms [38,39], its reduction with age creates an environment conducive to Candida colonization and subsequent infection.

Detailed Comments

Journal JoF (ISSN 2309-608X)

Manuscript ID jof-3749964

Type Article

Title Oral Candidiasis Associated with Aging and Salivary Hypofunction in Stomatitis Patients

Authors Yeon-Hee Lee * , Solsol Seo , Tae-Seok Kim , Sang-woo Lee *

Abstract :  There are several repetitions in the summary, 12 times "oral candidiasis" 8 times "stomatitis" which complicates reading. The distinction between stomatitis and oral candidiasis should be made at the outset in the abstract.

Response: This comment overlaps with the one previously raised regarding repetitions in the abstract. We kindly refer you to our response to that comment. We believe that we have sufficiently addressed your concerns by clarifying the distinction between stomatitis and oral candidiasis at the beginning of the abstract and by revising the abstract to improve clarity and avoid redundancy. Thank you again for your valuable feedback.

 Line 154: The identification and quantification of C albicans by culture alone currently remains subjective. Other fungi, bacteria, and viruses may be involved. At what quantitative threshold is C albicans considered pathogenic?

Response: Thank you for your insightful comment. We fully agree with your concern regarding the subjectivity of identifying Candida albicans based solely on culture morphology and the possible involvement of other fungi, bacteria, or viruses.

At our institution, the identification of Candida albicans is primarily based on characteristic colony morphology and color on Sabouraud dextrose agar, which is a widely accepted method in clinical practice. However, when the colony morphology or color was atypical, we conducted additional species identification using MALDI-TOF MS (Microflex, Bruker Daltonics), which provides highly reliable differentiation between Candida species. According to our institutional data, over 95% of oral isolates in stomatitis patients are confirmed as C. albicans through this approach. Regarding the pathogenic threshold, while there is no universally established quantitative cutoff for C. albicans colony counts in oral samples, colonies classified as "few" or more in culture were considered clinically meaningful in this study. This criterion aligns with routine clinical diagnostic practice at our institution. We have clarified this in the revised manuscript. Additionally, we acknowledge that other microorganisms could contribute to stomatitis; however, this study focused on Candida albicans due to its predominant prevalence and relevance to our hypothesis regarding salivary flow and aging. We have added this clarification to the revised Discussion section.

We appreciate your valuable suggestion and have incorporated these clarifications into the manuscript to address this point more clearly.

 Line 171:  It is important to note that prosthetic stomatitis is mostly painless. It is only a clinical examination that allows this pathology to be diagnosed, confirmed by a mycological analysis.

Response: Thank you for your valuable comment. We fully agree that prosthetic stomatitis is often painless and typically requires a clinical examination, alongside mycological analysis, for accurate diagnosis. We have added a statement in the revised Discussion to clarify that our findings, which include salivary flow rate and aging as predictors of oral candidiasis, may not directly apply to painless denture stomatitis. In such cases, diagnosis relies primarily on clinical and microbiological assessments rather than subjective symptoms. We appreciate your helpful suggestion.

We have revised the first paragraph of the Discussion section to reflect your valuable suggestion, as follows:

Approximately one-third of the patients with stomatitis had oral candidiasis. Patients with oral candidiasis were significantly older than those without candidiasis. The unstimulated and stimulated salivary flow rates were significantly lower in patients with oral candidiasis. The proportion of patients with xerostomia, based on both unstimulated and stimulated salivary flow rates, was significantly higher in the oral candidiasis group. Among the predictors assessed (age, UFR, and SFR), UFR showed the highest predictive accuracy for oral candidiasis. In the generalized linear model, xerostomia defined by UFR was identified as a significant predictor. Additionally, with increasing age, the occurrence of xerostomia based on UFR and oral candidiasis also increased. Contrary to our expectations, age alone was not a significant factor for oral candidiasis, and there was no difference in subjective pain intensity between patients with and without candidiasis. In addition, denture stomatitis is often asymptomatic and cannot be diagnosed based on subjective symptoms alone [20], highlighting the importance of objective assessments such as clinical examination, salivary flow measurements, and mycological testing for identifying oral candidiasis, particularly in denture wearers. Taken together, these findings underscore the importance of considering salivary gland function, aging, and systemic factors in the evaluation and management of oral candidiasis and stomatitis.

 Line 246: To enhance the value of this descriptive study, it would have been desirable to have more details on the number of teeth present, the presence of a removable or fixed prosthesis, caries or periodontal disease.

Response: Thank you for your valuable comment. Unfortunately, detailed data on the number of teeth, the presence of removable or fixed prostheses, dental caries, and periodontal disease were not available in this study. Due to the nature of our study design, there were limitations in collecting additional oral health data retrospectively. This study focused primarily on the relationship between oral candidiasis, salivary flow rate, aging, and xerostomia in patients with stomatitis, rather than on detailed dental status.

However, we have acknowledged this as a limitation in the revised Discussion section and noted that future studies should consider these factors to provide a more comprehensive understanding of the relationship between oral environmental conditions and oral candidiasis.

Thank you again for your thoughtful suggestion, which has helped improve the clarity of our study.

Despite the strengths of this study, several limitations should be noted. The uneven distribution of patients across different age groups, despite the large sample size of over 250 participants, may have introduced bias in age-related analyses. In addition, data on oral health status, such as the number of teeth, presence of prostheses, caries, and periodontal disease, were not collected in this study. Future studies…

 Line 335 :   This sentence requires a bibliographic reference. In these conditions, do you consider that oral candidiasis is part of stomatitis? 

Response: Thank you for your valuable comment. Oral candidiasis is considered a subset of stomatitis as it is one of the representative inflammatory conditions affecting the oral mucosa. Thank you again for this helpful suggestion.

Line 409 : addition to undisinfected prostheses, there are unsuitable, old removable appliances and resinous materials that facilitate microbial colonization. 

Response: We agree with your opinion. We have added this point to the revised manuscript as follows:

We focused on the relationship between the growth of Candida albicans and the presence of oral candidiasis in patients with stomatitis. However, there is a significant lack of research specifically addressing the role of C. albicans in the context of stomatitis. Oral candidiasis is a common opportunistic infection of the oral cavity caused by the overgrowth of Candida species, with C. albicans being the most prevalent [30]. Several local factors contribute to the development of oral candidiasis, including impaired salivary gland function, xerostomia, and poor hygiene of dental prostheses. In addition, unsuitable or old removable appliances and resin-based materials can further facilitate microbial colonization, creating an environment conducive to fungal overgrowth [31]. The tongue is also considered a major site for stomatitis-related symptoms, regardless of the presence of oral candidiasis, as it provides a favorable habitat for microorganisms [32]. Accumulated metabolic debris, such as food particles and desquamated epithelial cells, along with environmental factors such as humidity and temperature, further promote microbial proliferation on the tongue surface [33]. Beyond these local factors, systemic conditions also play a critical role. Antibiotic use, certain medications, aging, malnutrition, and immunosuppression are well-known systemic contributors to the development of oral candidiasis [34].

Conclusion: This statement: "Furthermore, oral candidiasis occurred significantly more frequently with age in patients with stomatitis." should be modulated because it is not applicable to denture stomatitis. Indeed, the epidemiology of denture stomatitis lists acute symptoms evident in young patients compared to older patients. Heightened immune defenses, poor hygiene, smoking, diet  … provide one explanation. In addition, fibrosis of the mucosal connective tissue in older patients represents a protection against denture covering.

Response: Accordingly, the Conclusion has been revised as follows:

Conclusion

A notable strength of this study is the scientific demonstration that oral candidiasis in patients with stomatitis has significant clinical implications. In patients with stomatitis, oral candidiasis was associated with decreased unstimulated salivary flow rate and increasing age. Specifically, the cut-off value for predicting oral candidiasis based on unstimulated salivary flow was 0.335 mL/min. Additionally, the prevalence of oral candidiasis increased significantly with age, with a cut-off value of 64.50 years for age. However, it should be noted that this age-related trend may not apply uniformly to all types of stomatitis. In particular, the epidemiology of denture stomatitis has been reported to differ, with acute symptoms often observed in younger individuals, while mucosal fibrosis in older patients may offer some protective effect against fungal colonization. Therefore, the association between aging and oral candidiasis should be interpreted in the context of salivary function and other contributing factors, rather than age alone. Another strength of this study is the validation of our hypotheses across a broad age range, from teenagers to individuals in their 80s. Expanding on this study, utilizing advanced oral microbiome analyses to identify specific profiles of stomatitis patients and microorganisms related to Candida albicans will be instrumental in furthering our understanding of this disease.

Add recent reference like…

Tamai, R.; Kiyoura, Y. Candida Infections: The Role of Saliva in Oral Health—A Narrative Review. Microorganisms 2025, 13, 717. https://doi.org/10.3390/ microorganisms13040717

Response: We have added the recommended reference as well as other relevant and up-to-date literature to further support and strengthen our study. We sincerely appreciate your valuable suggestion and contribution to enhancing the quality of our work.

Thank you again for your kind and insightful feedback.

Reviewer 2 Report

The work presented by Lee et al. demonstrates that oral candidiasis is associated with decreased salivary flow and increasing age in patients with stomatitis. The study is interesting, but presents some inconsistencies:

  1. In the abstract, the conclusion is not clearly compatible with the conclusion presented in the manuscript. It should be worded so that it is consistent with what is in the manuscript.
  2. L145-146 The statement “From the second week onward, the intervals between observations were gradually extended to monitor the growth of Candida albicans.” should be worded appropriately, as they imply that only albicans was isolated from the oral cavity or that the culture medium was selective for C. albicans. How do you know at this point that it is C. albicans?
  3. The authors mention that the yeasts were identified by PCR, but they don't show the results. They only identified albicans, but other species are present in the oral cavity. Why is this? This result should be discussed.
  4. Figure 1. How do you know it's albicans? According to the methodology description, you're referring to Figure 1 before you've identified the species.

L114-115 The statement: “A total of 259 patients (58 men and 201 women; mean age: 59.77 ± 15.93 years; range: 10–87 years) were included” should be included in the results.

Author Response

We sincerely appreciate your valuable comments and suggestions. We have carefully addressed your feedback to improve the clarity and quality of our manuscript. We believe that your insights have helped enhance our work.

All revised sections have been marked in red in the manuscript for your reference.

Thank you again for your time and effort in reviewing our paper.

Major comments

The work presented by Lee et al. demonstrates that oral candidiasis is associated with decreased salivary flow and increasing age in patients with stomatitis. The study is interesting, but presents some inconsistencies:

  1. In the abstract, the conclusion is not clearly compatible with the conclusion presented in the manuscript. It should be worded so that it is consistent with what is in the manuscript.

Response:

Thank you very much for your valuable comments.

We have revised the conclusion to improve clarity, as follows:

Before revision:

Although a decrease in salivary flow rate and aging were associated with the occurrence of oral candidiasis in patients with stomatitis, these factors alone did not result in high predictive accuracy.

After revision:

In patients with stomatitis, oral candidiasis was significantly associated with decreased unstimulated salivary flow rather than aging itself. Among the evaluated factors, decreased UFR may serve as a useful clinical indicator for predicting oral candidiasis, particularly in elderly individuals.

We hope this revision better clarifies the key conclusion of our study in line with your suggestion.

  1. L145-146 The statement “From the second week onward, the intervals between observations were gradually extended to monitor the growth of Candida albicans.” should be worded appropriately, as they imply that only albicanswas isolated from the oral cavity or that the culture medium was selective for  albicans. How do you know at this point that it is C. albicans?

Response: We sincerely appreciate your thoughtful comments. Our response to your suggestion is as follows.

Identification of Non-Candida albicans Species

At Kyung Hee University Medical Center, C. albicans is typically identified based on the characteristic morphology and color of colonies during the culture process in the Department of Laboratory Medicine. In most cases, colonies are correctly presumed to be C. albicans through this morphological assessment.

Generally, the majority of colonies isolated from stomatitis patients are C. albicans. However, other Candida species, such as Candida tropicalis and/or Candida parapsilosis, may occasionally be detected either alone or together with C. albicans. These non-albicans species are found in fewer than 5% of all patients. Therefore, it was reasonable to assume that most colonies were C. albicans when conducting this study.

When colony morphology or color did not correspond to the typical appearance of C. albicans, additional species identification was performed using Microflex: MALDI-TOF MS (Matrix-Assisted Laser Desorption Ionization Time-of-Flight Mass Spectrometry) to differentiate among Candida species.

  • Microflex: MALDI-TOF MS Process
  • Inoculation of the target plate with the cultured colony
  • Treatment with formic acid
  • Application of matrix solution
  • Ionization via laser beam
  • Measurement of ion flight time to the detector
  • Analysis of mass-to-charge ratio (m/z)
  • Differentiation based on species-specific ion patterns

  • Interpretation of Results: Score Value

Color

Score Value Range

Interpretation

Green

2.00 – 3.00

High-confidence identification

Yellow

1.70 – 1.99

Low-confidence identification

Red

0.00 – 1.69

No organism identification possible

  1. The authors mention that the yeasts were identified by PCR, but they don't show the results. They only identified albicans, but other species are present in the oral cavity. Why is this? This result should be discussed.

Response: Thank you for your valuable comment. As noted, PCR was used solely to confirm the presence of Candida albicans. Therefore, no additional information regarding other Candida species was obtained through this method.

We appreciate your thoughtful comment, and we will update the manuscript accordingly along with the revisions mentioned in our previous responses.

  1. Figure 1. How do you know it's albicans? According to the methodology description, you're referring to Figure 1 before you've identified the species.

Response: Thank you for your valuable comment. At Kyung Hee University Medical Center, Candida albicans is initially identified in the Department of Laboratory Medicine based on characteristic colony morphology and color observed on the culture medium. In routine clinical practice, these features are used for the presumptive identification of C. albicans, which is widely accepted in the diagnostic process.

In addition, according to our institutional experience, in more than 95% of oral samples, colonies that grow on the culture medium are ultimately identified as C. albicans through MALDI-TOF MS analysis. Therefore, it was reasonable to describe the colonies as C. albicans in Figure 1 based on this established clinical practice and prior knowledge. When colony morphology or color did not correspond to the typical appearance of C. albicans, additional species identification was performed using Microflex MALDI-TOF MS to confirm the species. We have revised the related description in the manuscript to clarify this point.

According to our institutional experience, in over 95% of oral samples, colonies that grow on culture media are ultimately confirmed as Candida albicans through matrix-assisted laser desorption ionization time-of-flight mass spectrometry analysis. In rare cases (fewer than 5% of patients), Candida tropicalis, Candida parapsilosis, Candida glabrata, or Candida krusei may be detected, either alone or in combination with other species of the Candida genus. Based on this evidence, colonies exhibiting typical morphological features were presumed to be C. albicans in this study.

Detailed comments

  1. L114-115 The statement: “A total of 259 patients (58 men and 201 women; mean age: 59.77 ± 15.93 years; range: 10–87 years) were included” should be included in the results.

Response: Thank you for your careful comment. We have moved the relevant content to the Results (Demographics) section for better clarity.
Thank you again for your helpful suggest

Round 2

Reviewer 2 Report

The modifications improved the presentation of the work, thank you.

The modifications improved the presentation of the work, thank you.